# The Role of COVID-19 in the Death of SARS-CoV-2–Positive Patients: A Study Based on Death Certificates

**DOI:** 10.3390/jcm9113459

**Published:** 2020-10-27

**Authors:** Francesco Grippo, Simone Navarra, Chiara Orsi, Valerio Manno, Enrico Grande, Roberta Crialesi, Luisa Frova, Stefano Marchetti, Marilena Pappagallo, Silvia Simeoni, Lucilla Di Pasquale, Annamaria Carinci, Chiara Donfrancesco, Cinzia Lo Noce, Luigi Palmieri, Graziano Onder, Giada Minelli

**Affiliations:** 1Division of Integrated Systems for Health, Social Assistance and Welfare, Italian National Institute of Statistics, 00184 Rome, Italy; frgrippo@istat.it (F.G.); simone.navarra@istat.it (S.N.); chiara.orsi@istat.it (C.O.); grande@istat.it (E.G.); crialesi@istat.it (R.C.); frova@istat.it (L.F.); stmarche@istat.it (S.M.); pappagal@istat.it (M.P.); silvia.simeoni@istat.it (S.S.); 2Statistical Service, Istituto Superiore di Sanità, 00161 Rome, Italy; valerio.manno@iss.it (V.M.); lucilla.dipasquale@iss.it (L.D.P.); annamaria.carinci@iss.it (A.C.); 3Department of Cardiovascular, Endocrine-Metabolic Diseases and Ageing, Istituto Superiore di Sanità, 00161 Rome, Italy; chiara.donfrancesco@iss.it (C.D.); cinzia.lonoce@iss.it (C.L.N.); luigi.palmieri@iss.it (L.P.)

**Keywords:** death certificates, COVID-19, comorbidities

## Abstract

**Background:** Death certificates are considered the most reliable source of information to compare cause-specific mortality across countries. The aim of the present study was to examine death certificates of persons who tested positive for severe acute respiratory syndrome coronavirus 2 (SARS-CoV-2) to (a) quantify the number of deaths directly caused by coronavirus 2019 (COVID-19); (b) estimate the most common complications leading to death; and (c) identify the most common comorbidities. **Methods:** Death certificates of persons who tested positive for SARS-CoV-2 provided to the National Surveillance system were coded according to the 10th edition of the International Classification of Diseases. Deaths due to COVID-19 were defined as those in which COVID-19 was the underlying cause of death. Complications were defined as those conditions reported as originating from COVID-19, and comorbidities were conditions independent of COVID-19. **Results:** A total of 5311 death certificates of persons dying in March through May 2020 were analysed (16.7% of total deaths). COVID-19 was the underlying cause of death in 88% of cases. Pneumonia and respiratory failure were the most common complications, being identified in 78% and 54% of certificates, respectively. Other complications, including shock, respiratory distress and pulmonary oedema, and heart complications demonstrated a low prevalence, but they were more commonly observed in the 30–59 years age group. Comorbidities were reported in 72% of certificates, with little variation by age and gender. The most common comorbidities were hypertensive heart disease, diabetes, ischaemic heart disease, and neoplasms. Neoplasms and obesity were the main comorbidities among younger people. **Discussion:** In most persons dying after testing positive for SARS-CoV-2, COVID-19 was the cause directly leading to death. In a large proportion of death certificates, no comorbidities were reported, suggesting that this condition can be fatal in healthy persons. Respiratory complications were common, but non-respiratory complications were also observed.

## 1. Introduction

Italy was the first country in Europe to be affected by the coronavirus 2019 (COVID-19) pandemic. The first confirmed case was reported at the end of January 2020, and a case-based surveillance system was established on 27 February 2020 [1]. The system collects data regarding all laboratory-confirmed cases of COVID-19, as well as deaths associated with COVID-19, which are defined as deaths in patients confirmed to have tested positive for severe acute respiratory syndrome coronavirus 2 (SARS-CoV-2). At the beginning of the pandemic, the lethality of COVID-19 in Italy was estimated to be greater than that in other countries, partly because of methods used to estimate and define deaths related to COVID-19 [2]. In Italy, all deaths in patients who tested positive for SARS-CoV-2 are counted as COVID-19–related by a national surveillance system. However, because different causes of death can occur in SARS-CoV-2–positive patients, this approach can lead to an overestimate of COVID-19-related deaths.

In patients infected by SARS-CoV-2, death can occur as the consequence of various complications, including pneumonia, acute respiratory distress syndrome, shock, and heart disease. In addition, the presence of comorbidities, and hypertension, diabetes, cardiovascular diseases, and chronic respiratory conditions in particular, can increase vulnerability to the development of these complications [3,4]. Death certificates are considered the most reliable source of information to compare cause-specific mortality across countries. However, limited data are available on the complications and comorbidities reported on death certificates of patients who presented with COVID-19. Some analyses of conditions existing prior to contracting COVID-19 were conducted by the UK Office of National Statistic [5].

The aim of the present study was to examine death certificates of persons who tested positive for SARS-CoV-2 to (a) quantify the number of deaths directly caused by COVID-19; (b) estimate the most common complications leading to death; and (c) assess the most common comorbidities.

## 2. Experimental Section

The surveillance system managed by the Italian National Institute of Health (Istituto Superiore di Sanità—ISS) collects information on all SARS-CoV-2-positive individuals throughout the country [1,2]. Data on all positive cases were obtained from all Italian regions and autonomous provinces. All deaths of patients who tested positive for SARS-CoV-2 through reverse transcription polymerase chain reaction amplification, independently from pre-existing diseases that may have caused or contributed to death, were tracked by the surveillance system.

Regions and autonomous provinces were also asked to supply copies of ISS death certificates of patients who died after testing positive for SARS-CoV-2. A joint group of researchers from the ISS and Italian National Institute of Statistics (Istat) was established to analyse these certificates.

Causes of death were classified according to provisions of the tenth edition of the International Classification of Diseases (ICD-10) [6] and the most recent guidelines issued by World Health Organisation on COVID-19–related deaths [7]. Iris software updated for COVID-19 (version 5.7 www.iris-institute.org) was used for cause-of-death coding. Approximately 80% of the death certificates were automatically coded by the software; the remaining certificates were coded by study researchers.

Death certificates completed by physicians comprise two parts. Part 1 contains the events leading to death, including the underlying cause (UC) that initiated the chain of events leading to death and the causal sequence of conditions caused by it. Part 2 includes other relevant conditions contributing to death but not part of the sequence reported in Part 1. According to ICD-10 coding procedures, a death due to COVID-19 is defined as a death for which COVID-19 was the UC.

Comorbidities were identified as conditions reported in Part 2 of the certificate or conditions different from COVID-19 that initiated chains of events reported in Part 1. To select comorbidities, a pre-existing validated algorithm developed for the study of multiple causes of death was used [8]. Details on the algorithm are provided in Appendix A. Complications of COVID-19 were identified among conditions reported in the sequence in Part 1 in which COVID-19 was the UC. Conditions that appeared due to COVID-19 more frequently than expected were considered complications of COVID-19. The expected frequency was calculated under the null hypothesis of random distribution of conditions within Part 1. Under the null hypothesis, and denoting *P* as the joint frequency in Part 1 of a given condition *c* and COVID-19 on different lines, we would expect that half of *P* times the condition *c* reported was due to COVID-19, and half *P* times the contrary would be the case. A comparison of the observed and expected frequencies was carried out using a chi-square test (further details are provided in Appendix B).

As of 28 May 2020, a total of 31,851 deaths in SARS-CoV-2–positive patients were reported in Italy. The 5311 death certificates received, analysed, and coded represented 16.7% of all deaths. These records were representative of the regional distribution of COVID-19 related deaths (see Appendix C).

### Ethical Issues

On 27 February 2020, the Italian Presidency of the Council of Ministers authorised the collection and scientific dissemination of data related to COVID-19 by the ISS and other public health institutions [9].

## 3. Results

Of the 5311 analysed death certificates, 3298 (62.1%) referred to men, and 2009 (37.8%) to women. Gender was not specified in four certificates. Age at death was specified in 5211 certificates: 305 (5.7%) deaths occurred in the age class 30–59 years, 1940 (36.5%) in the class 60–79 years, and 2966 (55.8%) in those 80 years of age or older. No death occurring in persons younger than 30 years was recorded in the study sample.

### 3.1. Deaths Due to COVID-19

Table 1 provides the distribution of underlying causes by sex and age in the study group. COVID-19 was the UC in 4691 of 5311 certificates (88.3%), with men representing a slightly higher percentage than women (89.4% versus 86.5%). The lowest proportion of deaths with COVID-19 as the UC was observed in the 30–59 age group (86.2%), and the highest was in the 60–79 age group (90.4%). The most common UCs besides COVID-19 included diseases of the circulatory system (4.4%), neoplasms (2.6%), and diseases of the respiratory system (1.1%); the latter nearly exclusively represented by chronic lower-respiratory conditions. The group aged 30–59 demonstrated the highest proportions of deaths with UCs of neoplasms (6.9%) and diabetes (2.0%), while circulatory system diseases were substantially more frequent in the group aged 80 or older (5.4%).

### 3.2. Comorbidities

No comorbidities besides COVID-19 were reported in 1469 death certificates (27.7%), with no sex differences and only a small variation among age groups (Table 2). In 3842 (72.3%) cases, at least one comorbidity was reported besides COVID-19: 32.0% reported one condition, 26.8% two comorbidities, and 13.6% three or more. The 30–59 age group showed a lower prevalence of deaths with two or more comorbidities and a slightly lower mean number of comorbidities compared with older age groups.

Overall, the most frequently reported comorbidities (Figure 1) were hypertensive heart diseases (18.4% of death certificates), diabetes mellitus (15.8%), ischaemic heart disease (13.0%), and neoplasms (12.4%). Conditions such as chronic lower-respiratory diseases, cerebrovascular diseases, dementia, and Alzheimer’s disease were also frequently reported (between 5% and 10% of cases). Differences by sex were observed only for hypertensive heart diseases, while dementia and Alzheimer’s disease were more frequent in women than in men (20.0% versus 17.5% and 11.3% versus 4.1%, respectively), while ischaemic heart disease was more frequent in men (15.3% versus 9.2%).

Prevalence of most conditions increased with age, with some exceptions. Diabetes demonstrated the highest prevalence in the group aged 60–79 years, while prevalence of neoplasms and obesity decreased with age; these latter conditions were the most common comorbidities in the 30–59 age group, being reported in 19.0% and 15.7%, respectively. The youngest age group also showed the highest prevalence of diseases of the nervous system (6.6%), chronic liver diseases (4.9%), and mental and behavioural disorders (5.6%).

### 3.3. Complications

Conditions frequently reported as complications of COVID-19 are shown in Figure 2 (all conditions analysed are reported in Appendix B, Table A2). Pneumonia and respiratory failure were the most common complications, being identified in 78% and 54% of certificates, respectively. Complications with an overall prevalence below 8% included shock, respiratory distress and pulmonary oedema, sepsis, heart complications such as heart failure, renal failure, encephalitis, pulmonary embolism, and acute myocardial infarction. For the most frequent conditions (pneumonia and respiratory failure), little variation was observed by sex and age, while some differences by age were observed for the least-frequent conditions. In the 30–59 age group, shock, respiratory distress, pulmonary oedema, and heart complications were more prevalent compared with older age groups.

## 4. Discussion

The present study analysed death certificates, which are widely considered the most reliable source of information to compare cause-specific mortality across countries. Results show that approximately 88% of deaths in SARS-COV2–positive patients were due to COVID-19 and their comorbidities, and cardiovascular conditions, diabetes, and neoplasms in particular, are common in patients dying with COVID-19. Respiratory conditions were the most common complications leading to death, and other non-respiratory complications were observed less frequently.

Consistent with previous studies, more deaths due to COVID-19 were reported for men than women. This finding contrasts with the case distribution documented in the Italian population, which displayed the opposite pattern (case distribution on August 10, 2020: 46.2% men, 53.8% women). Reasons for the higher number of deaths observed in men are not yet known [10].

Comorbidities were common in patients dying with COVID-19, suggesting that death was often the result of the concomitance and interaction of different chronic diseases with SARS-COV-2 infection. In line with other studies, we observed that patients dying with COVID-19 often present with hypertensive heart disease, ischaemic heart disease, cerebrovascular diseases, diabetes mellitus, chronic diseases of the low respiratory tract, or obesity [3,4].

The high prevalence of comorbidities in patients who died after a positive test for SARS-COV-2 led to a debate in the Italian media regarding the main causes of death in COVID-19 patients. In the presence of a high level of comorbidities (i.e., neoplasms, dementia, heart disease), COVID-19 may represent the final event leading to death, but not necessarily the main UC. The analysis of death certificates addressed this issue by showing that the vast majority of deaths occurring in patients who tested positively to SARS-COV2 were due to COVID-19.

The quality of the information reported by the certifying physicians may vary greatly: both under- and overreporting may occur [11] and causes of death are not generally confirmed by autopsies. Despite these limitations, data based on death certificates represent a primary source of epidemiologic information since they cover the overall population and the procedures for selecting and coding causes of death are highly standardized and internationally agreed leading to comparable and reliable figures.

Some relevant differences were evident in the clinical aspects of COVID-19 among younger adults. They suffer from slightly fewer comorbidities compared with older adults and often present with a single severe comorbidity (i.e., neoplasms), but they are more likely to die with non-respiratory complications. This finding suggests that, while in older adults, mortality is closely related to a pre-existing vulnerability due to presence of chronic conditions, COVID-19–related deaths in adults younger than 60 may be due to the development of complications.

Obesity is observed more commonly in younger adults than in older adults. It can increase the severity of respiratory complications by restricting ventilation, impeding diaphragm excursion, and reducing immune responses to viral infection [12]. This condition has been shown to increase the risk of mortality and admission to intensive care units in patients with COVID-19, and this association appears to be stronger in younger adults [13]. This observation confirms the obesity paradox hypothesis for older persons, for whom a higher body mass index is associated with superior outcomes.

## 5. Conclusions

Comorbidities were observed in a high percentage (72%) of deaths of people who tested positive for SARS-CoV-2. Nevertheless, COVID-19 was the cause directly leading to death in the vast majority (88%) of the cases. Moreover, in a relevant proportion of death certificates (28%), and also in younger people, no comorbidity besides COVID-19 was reported, suggesting that this condition can be fatal in healthy persons. The comorbidities most commonly associated with COVID-19 were hypertensive heart diseases, diabetes, ischaemic heart diseases, and obesity, with this latter condition found in younger people. A high percentage of people (78%) presented with pneumonia, but non-respiratory complications were also observed, particularly in younger age groups. This finding suggests that to reduce COVID-19 case fatality, treatment of respiratory conditions should be combined with appropriate management of comorbidities and strategies to prevent and mitigate the effects of non-respiratory complications.

## Figures and Tables

**Figure 1 jcm-09-03459-f001:**
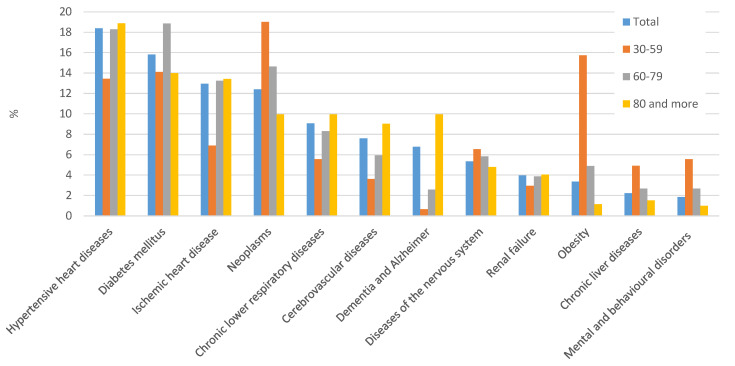
Distribution of the main comorbidities on death certificates, by age group.

**Figure 2 jcm-09-03459-f002:**
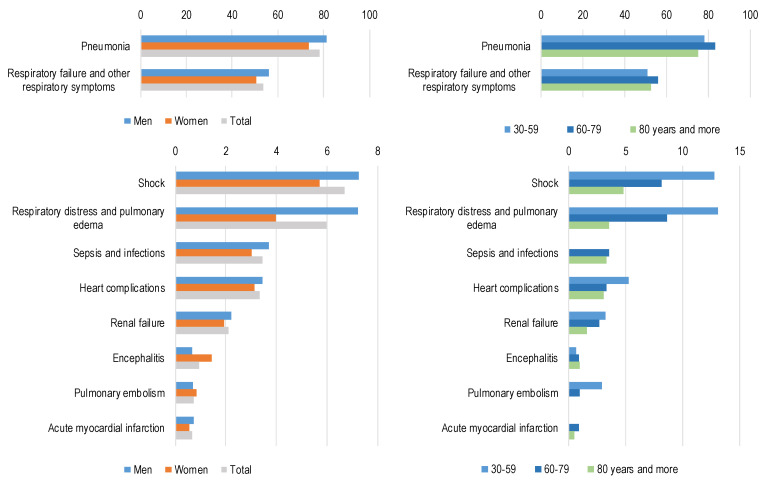
Main fatal complications of COVID-19 by gender and age group. Percentage of mention of the condition as complication of COVID-19 on total deaths.

**Table 1 jcm-09-03459-t001:** Underlying causes of death by gender and age group.

ICD-10	Underlying Cause of Death	Whole Sample		Gender				Age Group					
				Men		Women		30–59		60–79		80 and Older	
		N	%	N	%	N	%	N	%	N	%	N	%
U07.1	COVID-19	4691	88.3	2950	89.4	1737	86.5	263	86.2	1754	90.4	2590	87.3
A00-B99	Infectious and parasitic diseases	12	0.2	7	0.2	5	0.2	0	0.0	4	0.2	6	0.2
C00-D48	Neoplasms	137	2.6	87	2.6	50	2.5	21	6.9	54	2.8	57	1.9
E00-E99	Endocrine, nutritional and metabolic diseases	46	0.9	28	0.8	18	0.9	7	2.3	13	0.7	25	0.8
	of which												
*E10-E14*	*Diabetes*	*37*	*0.7*	*22*	*0.7*	*15*	*0.7*	*6*	*2.0*	*11*	*0.6*	*19*	*0.6*
F01-F03, G30	Dementia and Alzheimer	38	0.7	13	0.4	25	1.2	0	0.0	3	0.2	35	1.2
G00-H99	Diseases of the nervous system (excluding Alzheimer)	15	0.3	12	0.4	3	0.1	0	0.0	8	0.4	7	0.2
I00-I99	Diseases of the circulatory system	235	4.4	131	4.0	104	5.2	8	2.6	61	3.1	161	5.4
	of which												
*I10-I15*	*Hypertensive diseases*	*40*	*0.8*	*16*	*0.5*	*24*	*1.2*	*1*	*0.3*	*8*	*0.4*	*28*	*0.9*
*I20-I25*	*Ischaemic heart diseases*	*87*	*1.6*	*58*	*1.8*	*29*	*1.4*	*1*	*0.3*	*33*	*1.7*	*52*	*1.8*
*I48*	*Atrial fibrillation*	*12*	*0.2*	*5*	*0.2*	*7*	*0.3*	*0*	*0.0*	*2*	*0.1*	*10*	*0.3*
*I60-I69*	*Cerebrovascular diseases*	*55*	*1.0*	*29*	*0.9*	*26*	*1.3*	*2*	*0.7*	*11*	*0.6*	*41*	*1.4*
J00-J99	Diseases of the respiratory system	58	1.1	33	1.0	25	1.2	0	0.0	18	0.9	39	1.3
	of which												
*J40-J47*	*Chronic lower-respiratory diseases*	*55*	*1.0*	*33*	*1.0*	*22*	*1.1*	*0*	*0.0*	*17*	*0.9*	*37*	*1.2*
K00-K99	Diseases of the digestive system	36	0.7	18	0.5	18	0.9	2	0.7	15	0.8	17	0.6
S00-T98	External causes of death	26	0.5	12	0.4	14	0.7	1	0.3	6	0.3	18	0.6
	Other causes	28	0.6	13	0.4	15	0.7	4	1.3	9	0.5	15	0.5
	Total	5311	100.0	3298	100.0	2009	100.0	305	100.0	1940	100.0	2966	100.0

**Table 2 jcm-09-03459-t002:** Number of comorbidities by gender and age group.

Number of Comorbidities Besides COVID-19	Total	Gender	Age Group
Men	Women	30–59	60–79	80 and More
N	%	N	%	N	%	N	%	N	%	N	%
None	1469	27.7	911	27.6	554	27.6	80	26.2	550	28.4	819	27.6
at least 1	3842	72.3	2387	72.4	1455	72.4	225	73.8	1390	71.6	2147	72.4
of which:												
*1*	*1699*	*32.0*	*1059*	*32.1*	*640*	*31.9*	*124*	*40.7*	*621*	*32.0*	*920*	*31.0*
*2*	*1422*	*26.8*	*866*	*26.3*	*556*	*27.7*	*66*	*21.6*	*494*	*25.5*	*841*	*28.4*
*3 or more*	*721*	*13.6*	*462*	*14.0*	*259*	*12.9*	*35*	*11.5*	275	*14.2*	386	*13.0*
Total	5311	100.0	3298	100.0	2009	100.0	305	100.0	1940	100.0	2966	100.0
Mean number	1.3		1.3		1.3		1.2		1.3		1.3

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
