# Peer review of "The Role of COVID-19 in the Death of SARS-CoV-2–Positive Patients: A Study Based on Death Certificates"

_jcm, 2020, doi:10.3390/jcm9113459_

Round 1

Reviewer 1 Report

I am glad I had the opportunity to review this article since it addresses a vital entity. Although I get the feeling that the discussion could include more references, I understand that the authors wanted their work to be short and concise. 

Minor comments:

I am wondering if there were no cases of death below the age of 30 years?

Please be concise about dates (both are acceptable, but it is best to choose one style). Month Date or Date Month (for example, 104, 109).

Please change "particular" into "particularly" (212).

Figure 1 - please add a legend to the yellow bar (I suppose it refers to the group aged over 80 years).

Line 180 - The statement is correct; however, it is worth considering following study Agrawal H, Das N, Nathani S, Saha S, Saini S, Kakar SS, Roy P. An Assessment on Impact of COVID-19 Infection in a Gender Specific Manner. Stem Cell Rev Rep. 2020 Oct 7. doi: 10.1007/s12015-020-10048-z. Epub ahead of print. PMID: 33029768.

Please note that the author contributions should be corrected (delete "the following statement...").

Reviewer 2 Report

In this study, Grippo et al. use deaths certificates of persons who died after positive SARS-CoV-2 tests, to quantify the exact number of deaths cause by COVID-19, to analyze the most common complications leading to death, and to investigate the most common comorbidities. The Italian National Institute of Health manages a remarkable surveillance system to collect data on all COVID-19 patients nation-wide, and the researchers access all COVID-19 deaths reported through this system. Thus, they study an impressive number of more than 5000 death certificates of persons who died between March and May 2020 throughout Italy.

The paper is very well written and structured. The results are relevant and provide a useful and systematic summary and analysis of epidemiological data, the number and cause of deaths of COVID-19 patients as well as complications and comorbidities. The wide readership can draw various useful conclusions and learn a lot from this data. The paper is definitely of great importance.

Major points

  1. During the time period mentioned above, more than 30,000 deaths were reported in Italy. The authors need to explain why they included only about 5000 cases in their study, and how they selected these deaths certificates / patients.
  2. Discussion:

"The high prevalence of comorbidities in patients who died after a positive test for SARS-COV-2 led to a debate in the Italian media regarding the main causes of death in COVID-19 patients. In the presence of a high level of comorbidities (i.e., neoplasms, dementia, heart disease), COVID-19 may represent the final event leading to death, but not necessarily the main UC. The analysis of death certificates addressed this issue by showing that the vast majority of deaths occurring in patients who tested positively to SARS-COV2 were due to COVID-19."

In my opinion, you cannot make this conclusion without further discussion. Of course, death certificates are considered the most reliable source of information on cause-specific mortality. However, I guess that – if at all – only a minority of the patients in this study had a post mortem examination (autopsy) to analyze the cause of death, complications and comorbidities, respectively. And only the results of a post mortem examination can be considered fully reliable. I understand that the clinician who fills out the death certificate is right most of the time, but not always. This limitation should be discussed. It would be also good to mention if autopsies were performed and if the pathologists came to the same conclusions, namely that the cause of death was COVID-19 in the vast majority of cases.

On the other hand I understand, that this would be a separate study. Therefore I do not expect the researchers to access potential autopsy data easily.

Minor points

Figure 1

Labelling includes (...) This is not self-explanatory, and I recommend to include either full text or appropriate abbreviations and explain those in the legend.
